# Classroom-Situated Willingness to Communicate: Student Teachers of EFL in Spain

José Luis Estrada-Chichón [1], Francisco Zayas-Martínez [1] and Roberto Sánchez-Cabrero [2,*]

1 Department of Didactics of Language and Literature, Faculty of Education Sciences, University of Cádiz, 11519 Cadiz, Spain
2 Department of Evolutionary and Educational Psychology, Faculty of Teacher Training, Autonomous University of Madrid, 28049 Madrid, Spain
* Correspondence: roberto.sanchez@uam.es

**Abstract:** This exploratory mixed-methods research involves an intra-/intergroup replication design to analyse the classroom-situated willingness to communicate (WTC) in English foreign language (EFL) student teachers of early childhood education according to grouping, group member familiarity, and EFL proficiency. The novelty lies in the adaptation from face-to-face to virtual teaching while student teachers attended a didactics course at the University of Cádiz (Spain). Results show that there are no significant differences in WTC concerning teaching modality except for grouping. Conclusions imply that classroom-situated WTC is not affected by teaching modality when instruction aims at language acquisition by fostering oral communicative interaction.

**Keywords:** education; English teacher education; language acquisition; student teachers

## 1. Introduction

Early Childhood Education (henceforth ECE) degrees have been increasingly in demand in tertiary education in Spain over the last decade [1]. One of the specific characteristics these degrees share with some others within the field of education is their professional or qualifying nature. In other words, student teachers do not need to complement their undergraduate training with any subsequent postgraduate program.

So, at the University of Cádiz (henceforth UCA), Andalusia (Spain), the eight-terms curriculum of the degree in ECE distributes the 240 compulsory European Credit Transfer and Accumulation System (henceforth ECTS) points as follows: pedagogy courses (100 ECTS points); didactics courses (60 ECTS points); internships (44 ECTS points); specialization courses (30 ECTS points); and dissertation (6 ECTS points) [2]. Specialization courses allow students the opportunity to focus on a main area of interest and competence (inclusive education; social, behavioural, and emotional skills; art education; etc.). In this sense, the specialization course in Didactics of Foreign Language in Early Childhood Education (English) (henceforth Didactics) (6 ECTS points) is one of the five courses within the specialization of Linguistic and Literary Education (*educación lingüística y literaria*). Student teachers choose this course as an important milestone in their university education, also because of the current growing demand for English teachers in Spain [3].

Considering the administrative constraints and political circumstances of the primary and secondary education curricula in Spain, where the social demands for bilingual education have increased dramatically in the last two decades [4], Didactics is catching the attention of more and more students every year. As a result, the number of UCA student teachers enrolled in this course has increased from seven in 2018–2019, to 17 in 2019–2020, and to 20 in 2020–2021. Moreover, educational settings (including kindergartens and nursery schools) emphasize the importance of teaching English as a foreign language (henceforth EFL) or offering English–Spanish (bilingual) education as a sign of distinction and quality to give prestige to teaching [5–7].

Student teachers taking part in Didactics are aware that they must obtain an official certificate of B1 or B2 accreditation in any foreign language according to the Common European Framework of Reference for Languages (henceforth CEFR) [8] to graduate [9]. However, this course is not oriented towards linguistic objectives for obtaining any foreign language certificate, but mainly towards didactics training in foreign language teaching. Therefore, they should pay attention to the "emerging" (i.e., pre-programmed, in Chomskyan terms) language abilities of very young learners (0–6-year-old pupils) [10]. In the case presented in this study, along with foreign language teaching principles, student teachers explore how to use didactic resources as well as techniques and strategies to foster and enhance effective oral communication in EFL [11].

Being aware that achieving proficiency in EFL is neither part of nor a didactic objective of the course, one of the most revealing discoveries for student teachers is the use of EFL as the almost exclusive vehicle of communication [3,12]. Then, they must overcome their embarrassment and fear of speaking in EFL, not only to participate in practical activities but also to argue about didactic reasons for and against educational proposals, materials, and resources, teaching styles, etc. The vehicular use of EFL becomes a bonus for their professional development as teachers. Thus, pupils' access to comprehensible input depends to a large extent on (this type of) teacher training [13]. The almost total absence of written language, not even as an instructional model for language learning, represents a second, no less determinant impact of Didactics, where written literacy development is not a didactic objective.

Alongside the basic limitations for the treatment of EFL, this course provides student teachers with a relatively alternative methodological approach based mainly on the research principles of instructed second language acquisition (henceforth ISLA) [14–18]. In this respect, ISLA usually considers the cognitive (i.e., learner-centred) nature of language before its application for only communicative purposes [19]. This is how human language capacity helps expand the conceptual world of persons, which consequently involves the development of language acquisition [20], avoiding the application of pre-characterized linguistic behaviours connected to specific communicative situations [21].

Concerning the absence of written language and the need for motivational elements, a methodological approach to EFL based on ISLA should consider willingness to communicate (henceforth WTC) as a pivotal issue for initial diagnosis [22]. In other words, WTC represents "a learner's readiness to enter into discourse at a particular time with a specific person or persons using a L2" [23] (p. 547). WTC is used as a conceptual framework for measuring differences in oral fluency by ISLA-based research [24]. Previously, Derwing et al. (2008) [25] had already associated WTC with socio-affective factors and motivational issues, where anxiety derived from the contextual conditions of instruction and the relationship with other speakers can be considered [26]. Other researchers recommend reconsidering studies on fluency in a more holistic way, such as to include insights from all disciplines involved: psycholinguistics, psychology, applied linguistics and foreign language teachers [27]. In this sense, student teachers in Didactics can assess their WTC in EFL through systematic observation of the circumstances that may have a positive or negative effect on them. For this research, student teachers answered questions after each practical class on three different elements of analysis: (i) their WTC under different types of grouping; (ii) how WTC is conditioned by their relationships with other speakers; and (iii) how WTC is influenced by the linguistic competence (i.e., EFL competence) of other classmates [28]. Next, it was necessary to filter their evaluations through online discussion forums too so that they would become increasingly aware of the weight the three elements have in EFL acquisition.

The final contextual element has to do with the insistence throughout Didactics on an ISLA research-based role for student teachers to act as pedagogical mediators for authentic communication and as facilitators of EFL acquisition [29]. This takes place gradually, through carefully designed authentic stimuli as well as by providing enough room for individual exploration of the pupils' "language-making ability" [30] (p. 285). By the end of the course, student teachers should be aware that any foreign language (e.g., English, as it is the current context of analysis) should be acquired and not consciously learned [31]. In ECE classroom settings, the target language is used for ordinary classroom routines, among other activities.

Therefore, this study analyses student teachers' evaluations and first-hand perceptions about their WTC in EFL in a very specific context of study: The adaptation from face-to-face to virtual teaching within the specialization course Didactics at UCA (2019–2020) due to the spread of the COVID-19 disease and the lockdown enforced by the Spanish Government in all the territories of the country. This classroom-based research should make student teachers aware of their WTC and the specific circumstances that might positively or negatively affect it, which should also enable them to observe and/or consider how to deal with the WTC of their future pupils in terms of EFL development.

In this regard, WTC has been a paramount issue in recent studies and methodological proposals concerning the communicative approach (henceforth CA) to language teaching [32–34]. The CA prevails the communicative (i.e., information transfer) use of the target language over its cognitive (i.e., information processing and management) function [35]. So, WTC represents the starting point for the development of any communication sequence and thus also for the analysis of the significance it might have for EFL learning [36]. Even when simulated communicative exchanges take place [35], teaching models based on the CA implicitly assume that WTC depends on the (controllable and manipulated) elements of a given communicative act [37]. So, proposing a specific classroom setting that conditions in a natural way the possible communicative purpose of each participant can be relatively easy for any foreign language teacher and (once the functionally profitable linguistic material in that classroom setting is known) for the learners too.

However, in recent decades, there has been a significant improvement in the field of theoretical foundations underpinning methodological proposals for foreign language teaching [38]. These (implicitly or explicitly) claim for the importance of the cognitive nature of language and its process of acquisition [39,40] as central to the methodological proposal to be followed. For example, the *enseñanza adquisitiva* (acquisitive teaching) of foreign languages [41,42]; dual teaching–learning modality for the integration of content and language(s) such as content–language integrated learning (henceforth CLIL) [43,44]; and the large body of research on ISLA [14,15]. For the first time, it brings together theoretical concepts traditionally irreconcilable, such as language acquisition and methodological guidelines for foreign language teaching. This is an opportunity to consider some key concepts presented by Dörnyei and Ushioda (2011) [45]—such as "language attitudes", the "ideal L2 self" and the "should-be L2 self", among others conventionally associated with motivation—as elements that can be monitored and investigated in foreign language teaching practice, as suggested by Csizér [46].

WTC reaches then a different level, as learners' disposition to engage in a communicative act proposed by the foreign language teacher depends on cognitive elements independent of the communicative scenario itself [23]. This also includes the learner's assessment of his or her command of the target language associated with that specific scenario [22], i.e., the lexicon used for the treatment of a given topic and the linguistic structures necessary for the assumption of a specific role in it. To address these conditioning elements, scholars distinguish between situated WTC, on the one hand, and trait-like WTC, on the other. Situated WTC involves language-use opportunities, where a (higher or lower) predisposition towards communication depends on predictable and observable factors that must be pre-programmed by foreign language teachers for a specific purpose, such as group size, self-confidence, familiarity with interlocutors, and interlocutor in-

volvement [28], for example. Meanwhile, trait-like WTC refers to a "stable predisposition towards communication" [47] (p. 3).

Situated WTC elements linked to psychological factors such as perception of security, excitement, and responsibility [48] can be controlled for authentic communication purposes within a language-acquisition teaching context. This setting is claimed as classroom-situated WTC [49]. Here, the social and psychological characteristics of learners constantly interact with the changing conditions of the learning environment [50]. Therefore, classrooms represent a natural context for situated WTC research [47]. Learners' emotions—and not only consciously predictable trait-like WTC factors—can be considered as the most conditioning and sensitive factors for WTC [51,52]. Consequently, by placing this methodological proposal in the realm of cognitive-based approaches, it assumes the consideration of situated WTC, which is systematically embedded in the classroom as the only authentic setting for language acquisition.

## 2. Materials and Methods

### 2.1. Objectives and Research Hypotheses

The general objectives of this study are:

1. To determine to what extent the ISLA-based methodological approach has a positive effect on student teachers' WTC in terms of grouping, group member familiarity, and EFL proficiency.
2. To find out the extent to which student teachers' WTC is sustained when adapting the course to a remote setting, i.e., from face-to-face to virtual teaching. The following research hypotheses are posed:
3. An ISLA-based methodological approach can contribute to the consolidation of WTC without the need for simulated uses of EFL. So, WTC improves when:
   a. Student teachers work in pairs rather than in groups or with the whole class.
   b. Student teachers speak to someone they know well rather than to someone they know a little or to the teacher.
   c. The EFL proficiency of the interlocutor is lower than or equal to that the speaker's EFL proficiency.
4. The consolidation of student teachers' WTC throughout an ISLA-based methodological approach is sustained when course teaching modality changes from face-to-face to virtual.

### 2.2. Experimental Design

An exploratory mixed-methods research was conducted thoroughly, combining a repeated-measures design with an inter-group design for a single sample. The intergroup design multiplies the types of statistical error, although it drastically reduces the amount of error of each type. It is therefore considered a very robust experimental design [53]. In addition, it allows for a considerable reduction in the sample needed to reach valid statistical conclusions, as a small group has the statistical generalization potential of a group several times larger in simpler designs.

### 2.3. Context and Participants

The research context is the six 90-min practical classes of the optional 6-ECTS-point course Didactics. This course belongs to the specialization of Linguistic and Literary Education in the bachelor's degree of ECE at UCA. This course is taught in semester 6 (Year 3, second semester), so it coincided with the lockdown enforced by the Spanish Government in the country from 18 March 2020 onward due to the spread of the COVID-19 infectious disease. This implied an unexpected change in the course modality from face-to-face to virtual teaching (see Table 1).

In relation to the course contents, Block I (practical classes) focuses on "Developing the linguistic competence of the future teacher of English in pre-primary education." It includes the following sections: (i) classroom language and essential grammar in use and (ii) basic semantic fields to be addressed in early childhood education (colours, shapes, numbers, animals, etc.):

**Table 1.** Classroom research timeline including course teaching modality.

|  | Feb. 18 | Mar. 3 | Mar. 10 | Apr. 1 | Apr. 14 | Apr. 21 |
|---|---|---|---|---|---|---|
| Face-to-face | X | X | X |  |  |  |
| Virtual |  |  |  | X | X | X |

The study sample is made up of 15 female student teachers (18–20 years old). Sampling was performed by clustering, selecting all the student teachers attending Didactics in the academic year 2018–2019. Furthermore, the highest language (i.e., FL) proficiency level accredited among the student teachers is B1 according to the CEFR, although there is no language restriction to attending the course.

*2.4. Research Tools and Variables*

In order to collect the quantitative data for this research study, an adaptation of an online questionnaire [36,54] on the "dynamic nature of L2 willingness to communicate" [36] was used. The questionnaire was answered individually by each student teacher at the end of each (face-to-facer or virtual) class as for three main elements of analysis: grouping, group member familiarity, and EFL proficiency:

1.  How willing were you to communicate in English in class today?
2.  How willing are you to communicate in English after today's class?
3.  Grouping: pairs
4.  Grouping: groups
5.  Grouping: whole class
6.  Group member familiarity: someone I know very well
7.  Group member familiarity: someone I know a little
8.  Group member familiarity: teacher
9.  Interlocutor's EFL proficiency: same as mine
10. Interlocutor's EFL proficiency: higher than mine
11. Interlocutor's EFL proficiency: lower than mine
12. Self-evaluation of the class evaluate today's class
13. Open questions:
    a.  List the factors that helped increase your WTC in English
    b.  List the factors that helped to decrease your WTC in English
    c.  What activities helped increase your WTC in English?
    d.  What activities helped to decrease your WTC in English?
    e.  When did you feel more willing to communicate in English today?
    f.  When did you feel less willing to communicate in English today?

The questionnaire includes 12 self-assessment items (Q1–Q12) on a 1 (very bad)–5 (very good)-point Likert scale on different WTC-related issues within the ISLA-based methodological approach applied to the six practical classes, and six open questions. The questionnaire has excellent reliability (Table 2), as measured by Cronbach's Alpha for each class:

**Table 2.** Questionnaire reliability.

|  | Class | Cronbach's Alpha |
|---|---|---|
| 1 | Face-to-face | 0.895 |
| 2 | Face-to-face | 0.768 |
| 3 | Face-to-face | 0.833 |
| 4 | Virtual | 0.898 |
| 5 | Virtual | 0.866 |
| 6 | Virtual | 0.941 |

Before answering the questionnaire, all student teachers were informed of the research and received written informed consents. The research was conducted according to scientific and ethical standards, monitoring strict compliance with the 2013 Declaration of Helsinki on the Ethics of Research Involving Human Subjects [55].

The main variables evaluated in the present manuscript are described below:

1.  Classes using an ISLA-based methodological approach (ordinal intragroup variable): six classes where the evaluation instrument (questionnaire) was applied at the end.
2.  Teaching modality (nominal intergroup variable with two levels: face-to-face and virtual teaching). First, the three face-to-face classes were taught and then the three final classes were taught virtually.
3.  WTC (quantitative variable). It is measured through Q1 and Q2 of the questionnaire on a 1–5-point Likert scale.
4.  Grouping (nominal variable including three levels: (i) pairs, (ii) groups, and (iii) whole class). It is measured through items Q3, Q4, and Q5 of the questionnaire.
5.  Group member familiarity (nominal variable including three levels: (i) someone I know very well; (ii) someone I know a little; and (iii) teacher). It is measured through items Q6, Q7 and Q8 of the questionnaire.
6.  Interlocutor's EFL proficiency (nominal variable including three levels: (i) higher than mine; (ii) lower than mine; and (iii) same as mine). It is measured through items Q9, Q10, and Q11 of the questionnaire.
7.  Self-evaluation of each (face-to-face or virtual) class (ordinal variable). It is measured through item Q12 of the questionnaire.

On the other hand, main qualitative data were obtained through the student teachers' answers to twelve questions (see Appendix A) presented in an online discussion forum to illustrate the qualitative data collected through the questionnaire.

*2.5. Data Analysis*

Once the data had been collected, the following statistical analyses were carried out in relation to the research objectives:

1.  Evaluation of intrasubject consistency through the intraclass correlation coefficient. This analysis evaluates whether student teachers' responses showed consistency according to their unique interindividual characteristics.
2.  RGraphical analysis (i.e., Figures 1–4) of the evolution of the variables throughout the classes. The evolution of the WTC and each of the variables can then be observed visually.
3.  Analysis of the evolution of the variables throughout the classes using repeated measures ANOVA. It is evaluated whether there is a significant learning or unlearning effect over the classes and whether this effect, together with the change in teaching modality, alters the results achieved.
4.  Comparison of the effects of face-to-face versus virtual teaching through Student's *t*-test.

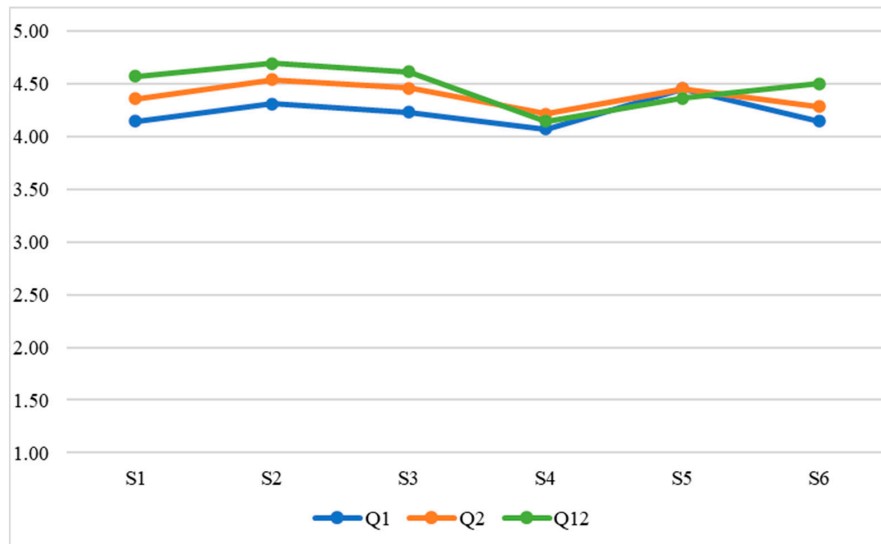

**Figure 1.** Evolution of WTC and self-evaluations.

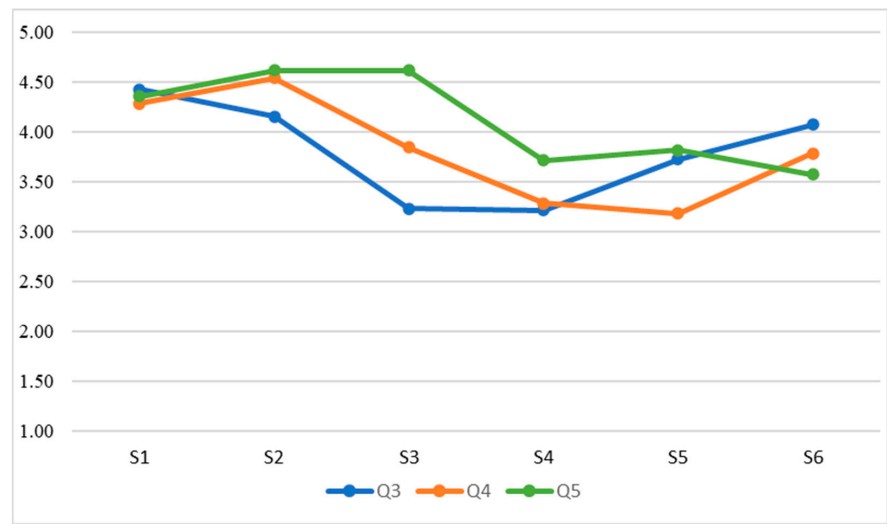

**Figure 2.** Grouping.

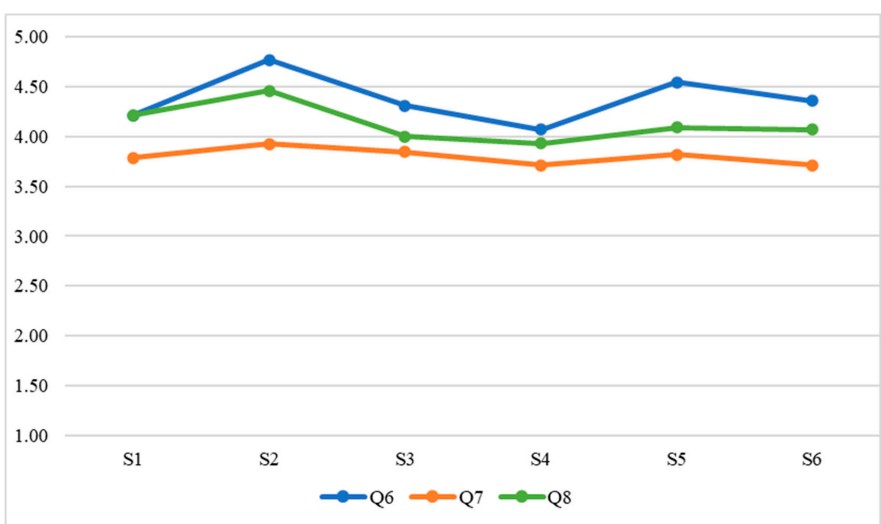

**Figure 3.** Group member familiarity.

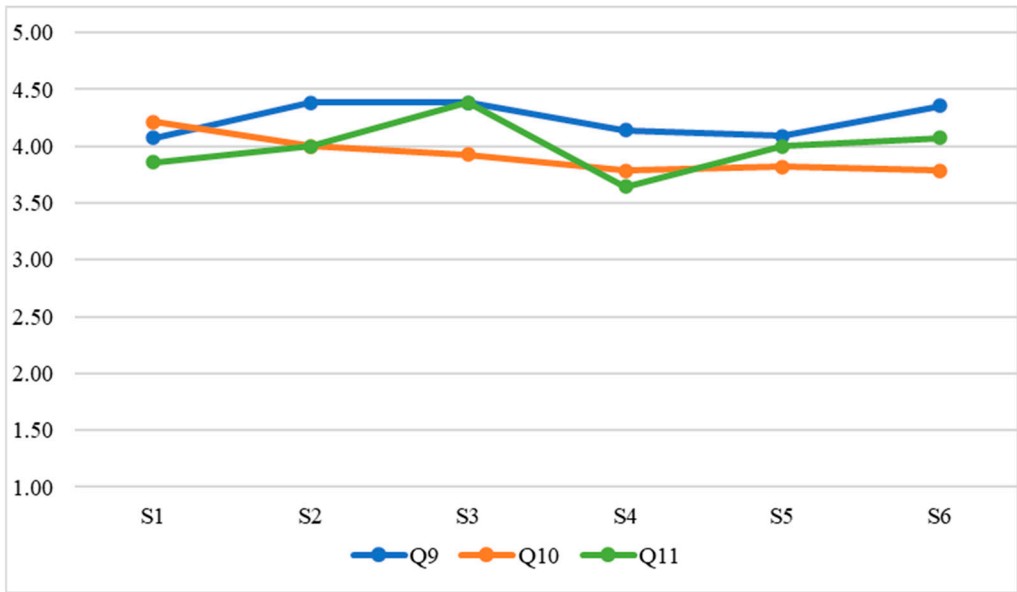

**Figure 4.** Interlocutor EFL proficiency.

## 3. Results

The first step is to analyse the consistency of the student teachers' answers throughout the six practical classes, since a break in this consistency will be a major reason that could explain the results of the subsequent analyses. Table 3 shows the analysis of intra-subject consistency for each question:

**Table 3.** Intraclass correlation coefficient (ICC).

|  | WTC | ICC | *p* Value |
|---|---|---|---|
| Q1 | How willing were you to communicate in English in class today? | 0.796 | 0.000 |
| Q2 | How willing are you to communicate in English after today's class? | 0.868 | 0.000 |
|  | Grouping | | |
| Q3 | Pairs | −0.054 | 0.656 |
| Q4 | Groups | 0.453 | 0.000 |
| Q5 | Whole class | 0.498 | 0.000 |
|  | Group member familiarity | | |
| Q6 | Someone I know very well | 0.230 | 0.028 |
| Q7 | Someone I know a little | 0.716 | 0.000 |
| Q8 | Teacher | 0.665 | 0.000 |
|  | Interlocutor's EFL proficiency | | |
| Q9 | Same as mine | 0.544 | 0.000 |
| Q10 | Higher than mine | 0.644 | 0.000 |
| Q11 | Lower than mine | 0.127 | 0.118 |
|  | Self-evaluation of the class | | |
| Q12 | Evaluate today's class | 0.458 | 0.000 |

As observed in Table 3, only two of the 12 items show low intrasubject consistency (Q3 and Q11). However, this is a very sensitive value to small changes in some student teachers, so the results should be treated with caution pending further analysis.

Concerning the first objective of the research study (to determine to what extent the ISLA-based methodological approach has a positive effect on ECE student teachers' WTC in terms of grouping, group member familiarity, and EFL proficiency), the development of the student teachers' evaluations for each element (grouping, group member familiarity, and EFL proficiency) is shown in Figures 1–4; i.e., WTC and self-evaluation of each class (Figure 1) as for grouping (Figure 2), group member familiarity (Figure 3), and interlocutor's EFL proficiency (Figure 4). In Figure 1, there is no visual evidence of learning or unlearning due to the change in course teaching modality, nor a clear differentiation between face-to-face and online teaching, which addresses the second objective of the study (to find out the extent to which student teachers' WTC is sustained when adapting the course to a remote setting, i.e., from face-to-face to virtual teaching).

In Figure 2, learning or unlearning due to the change in course teaching modality is not clearly visible in terms of grouping, although there is some differentiation between face-to-face and virtual teaching.

Figure 3 does not show any learning or unlearning due to the change in course teaching modality, nor a clear differentiation between face-to-face and virtual classes as for group member familiarity. However, it is noteworthy that, as familiarity between student teachers decreases, their self-evaluation marks also decrease:

In Figure 4, there is no visual evidence of learning or unlearning due to the change in course teaching modality in terms of interlocutor's EFL proficiency, nor a clear differentiation between face-to-face and virtual classes.

In order to confirm the statistical validity of the research study, an analysis of the evolution of the variables throughout the six classes should be carried out using repeated measures ANOVA. This test allows us to check whether there is a significant effect of learning or unlearning over the classes (objective 2) and whether there is any variable that interferes with the process (objective 1). Table 4 shows the multivariate analyses, using the Pillai trace as the contrast statistic, of the repeated measures ANOVA for each of the questionnaire items:

**Table 4.** Multivariate tests (Pillai's trace).

| | Effect | Value | F | Gl. Hyp. | Gl Error | *p* | Partial Eta Squared |
|---|---|---|---|---|---|---|---|
| Q1 | Differences between classes (intra-) | 0.429 | 0.562 | 4.000 | 3.000 | 0.710 | 0.429 |
| Q1 | Impact of the variable "course teaching modality" (inter-) over the evolution of the classes (intra-) | 0.041 | 0.408 | 2.000 | 19.000 | 0.670 | 0.041 |
| Q2 | Differences between classes (intra-) | 0.429 | 1.000 | 3.000 | 4.000 | 0.479 | 0.429 |
| Q2 | Impact of the variable "course teaching modality" (inter-) over the evolution of the classes (intra-) | 0.023 | 0.224 | 2.000 | 19.000 | 0.801 | 0.023 |
| Q3 | Differences between classes (intra-) | 0.848 | 2.225 | 5.000 | 2.000 | 0.339 | 0.848 |
| Q3 | Impact of the variable "course teaching modality" (inter-) over the evolution of the classes (intra-) | 0.391 | 6.090 | 2.000 | 19.000 | 0.009 | 0.391 |
| Q4 | Differences between classes (intra-) | 0.647 | 0.733 | 5.000 | 2.000 | 0.663 | 0.647 |

**Table 4.** *Cont.*

|  | Effect | Value | F | Gl. Hyp. | Gl Error | *p* | Partial Eta Squared |
|---|---|---|---|---|---|---|---|
| Q4 | Impact of the variable "course teaching modality" (inter-) over the evolution of the classes (intra-) | 0.286 | 3.802 | 2.000 | 19.000 | 0.041 | 0.286 |
| Q5 | Differences between classes (intra-) | 0.571 | 1.000 | 4.000 | 3.000 | 0.521 | 0.571 |
| Q5 | Impact of the variable "course teaching modality" (inter-) over the evolution of the classes (intra-) | 0.104 | 1.108 | 2.000 | 19.000 | 0.351 | 0.104 |
| Q6 | Differences between classes (intra-) | 0.571 | 1.000 | 4.000 | 3.000 | 0.521 | 0.571 |
| Q6 | Impact of the variable "course teaching modality" (inter-) over the evolution of the classes (intra-) | 0.323 | 4.532 | 2.000 | 19.000 | 0.025 | 0.323 |
| Q7 | Differences between classes (intra-) | 0.079 | 0.034 | 5.000 | 2.000 | 0.998 | 0.079 |
| Q7 | Impact of the variable "course teaching modality" (inter-) over the evolution of the classes (intra-) | 0.048 | 0.482 | 2.000 | 19.000 | 0.625 | 0.048 |
| Q8 | Differences between classes (intra-) | 0.714 | 1.000 | 5.000 | 2.000 | 0.569 | 0.714 |
| Q8 | Impact of the variable "course teaching modality" (inter-) over the evolution of the classes (intra-) | 0.318 | 4.426 | 2.000 | 19.000 | 0.026 | 0.318 |
| Q9 | Differences between classes (intra-) | 0.571 | 1.000 | 4.000 | 3.000 | 0.521 | 0.571 |
| Q9 | Impact of the variable "course teaching modality" (inter-) over the evolution of the classes (intra-) | 0.201 | 2.390 | 2.000 | 19.000 | 0.119 | 0.201 |
| Q10 | Differences between classes (intra-) | 0.714 | 1.000 | 5.000 | 2.000 | 0.569 | 0.714 |
| Q10 | Impact of the variable "course teaching modality" (inter-) over the evolution of the classes (intra-) | 0.074 | 0.757 | 2.000 | 19,000 | 0.483 | 0.074 |
| Q11 | Differences between classes (intra-) | 0.714 | 1.000 | 5.000 | 2.000 | 0.569 | 0.714 |
| Q11 | Impact of the variable "course teaching modality" (inter-) over the evolution of the classes (intra-) | 0.033 | 0.323 | 2.000 | 19.000 | 0.728 | 0.033 |
| Q12 | Differences between classes (intra-) | 0.714 | 1.875 | 4.000 | 3.000 | 0.316 | 0.714 |
| Q12 | Impact of the variable "course teaching modality" (inter-) over the evolution of the classes (intra-) | 0.070 | 0.716 | 2.000 | 19.000 | 0.501 | 0.070 |

Moreover, Table 5 shows how the evolution of the six classes does not modify the results regarding any of the elements. However, the confluence of evolution throughout the sessions (intra variable) together with the change in course teaching modality from the third class onwards does generate important differences in items Q3, Q4, Q6, and Q8. Finally, in order to provide a more detailed response to the second research objective, a comparison is made between averages using Student's t-test as the contrast statistic:

**Table 5.** Course teaching modality: Face-to-face vs. virtual teaching.

| | Item | *t* | Significance |
|---|---|---|---|
| Q1 | How willing were you to communicate in English in class today? | 0.176 | 0.861 |
| Q2 | How willing are you to communicate in English after today's class? | 0.832 | 0.413 |
| Q3 | Pairs | 1.399 | 0.173 |
| Q4 | Groups | 2.277 | 0.032 |
| Q5 | Whole class | 2.563 | 0.019 |
| Q6 | Someone I know very well | 1.087 | 0.289 |
| Q7 | Someone I know a little | 0.514 | 0.611 |
| Q8 | Teacher | 0.706 | 0.486 |
| Q9 | Same as mine | 0.640 | 0.527 |
| Q10 | Higher than mine | 0.877 | 0.390 |
| Q11 | Lower than mine | 1.103 | 0.282 |
| Q12 | Evaluate today's class | 1.453 | 0.158 |

As observed in Table 5, only items Q4 and Q5 reveal important differences comparing the averages of the first three face-to-face classes with the last three online classes. Thus, it can be confirmed that the change of course teaching modality affected WTC in relation to grouping regarding group and whole class work.

## 4. Discussion

The starting point of this study is classroom-situated WTC [49], where the characteristics that define it are changeable [28]: the type of grouping varies not only according to the didactic objective of the activity, but also according to the change of course teaching modality from face-to-face to virtual; group member familiarity also varies throughout the course (e.g., a student teacher does not know another student teacher in the first class but may know him or her in session 6); and the interlocutors' EFL proficiency is not the same throughout all classes (i.e., it depends on who the student teachers are working with in each class).

The results obtained from the student teachers' answers to the questionnaire confirm that an ISLA-based methodological approach may contribute to the consolidation of WTC (research hypothesis 1: an ISLA-based methodological approach can contribute to the consolidation of WTC without the need for simulated uses of EFL), which is hardly affected by the change of course teaching modality from face-to-face to virtual (research hypothesis 2: the consolidation of student teachers' WTC throughout an ISLA-based methodological approach is sustained when course teaching modality changes from face-to-face to virtual), as pointed out in the online discussion forum ("teaching in ECE can always be possible if there are enough resources"; student teacher 6, personal communication, May 2020), except in relation to group and whole class work, where a slight decrease of WTC is observed. Reviewing the contributions of the student teachers in the online discussion forum, it may be found that they consider as major obstacles the lack of dynamism concerning virtual teaching ("it is not possible to carry out collective games in which pupils can interact with each other"; student teacher 15, personal communication, May 2020), bearing in mind that other student teachers remained seated on the other side of the screen while virtual teaching (sessions 4–6) was taking place ("sitting in front of a computer is not as effective for them [pupils] as it is for grown-ups"; student teacher 10, personal communication, May 2020), together with the fact that interaction is always teacher–student ("I think that in ECE, learning a foreign language through virtual teaching is quite difficult, as children need to touch, see, manipulate, observe, have fun with their classmates, etc. and I don't think

they would like to be in front of a screen following the teacher's instructions"; student teacher 13, personal communication, May 2020) [56]. In addition, student teachers stand out the dependence on parental involvement ("it is time for families to share time with their children and they can embark on this 'journey' together"; student teacher 4, personal communication, May 2020), considering the technical difficulties of virtual teaching for children ("this could be carried out in virtual ECE classes, with the [PPT] slides, the microphone for them to use oral EFL, and the [virtual] rooms for them to talk to their classmates"; student teacher 11, personal communication, May 2020).

All types of grouping are slightly affected by the change of course modality, but while work in pairs or with the whole class has been positively valued by the student teachers from class 4 onward, the same happens for group work from class 5 onward, which answers research hypothesis 1.a (WTC improves when student teachers work in pairs rather than in groups or with the whole class). Perhaps one of the reasons for the positive evaluation of working in pairs is due to the privacy online teaching offers [57], where student teachers do not feel threatened by being judged by others. The explanations provided in the online discussion forum point to the lack of physical proximity ("face-to-face work is more dynamic and 'closer' –we see both good and bad reactions and it manages to create a more positive atmosphere–, while virtual learning is 'colder', and we are more afraid of doing or saying something wrong"; student teacher 5, personal communication, May 2020); to the lack of interaction ("what we do is to respond individually and work in pairs, we do not all work together"; student teacher 1, personal communication, May 2020); and, finally, to the pace of interactions ("the willingness of students to work in groups is less complicated when doing it face-to-face, since we can do more things, we see each other's faces, etc."; student teacher 13, personal communication, May 2020).

In general terms, a slightly higher WTC in EFL in face-to-face teaching is also confirmed by the student teachers for all types of grouping ("face-to-face classes allow for greater interaction in large groups, while in online classes we should make an effort, i.e., being responsible for ourselves and our learning. Sometimes we disconnect when it is not our turn to speak in a large group. On the contrary, we do not disconnect for a moment in face-to-face classes, we do not have distractions from cell phones because there are no Internet outages"; student teacher 2, personal communication, May 2020) and regardless of the degree of group member familiarity ("I consider that working with a person I know very well is just as good in person as virtually. Likewise, I think it is more difficult with someone I know less, as there is not enough trust between us"; student teacher 1, personal communication, May 2020), which addresses research hypothesis 1.b (WTC improves when student teachers speak to someone they know well rather than to someone they know a little or to the teacher), and the interlocutor's EFL proficiency ("I think it is more difficult to communicate virtually because, when doing it in person, we use many resources such as gestures, expressions, facial and body movements . . . ", student teacher 2; personal communication, May 2020), which refers to research hypothesis 1.c (WTC improves when the EFL proficiency of the interlocutor is lower than or equal to that the speaker's EFL proficiency of the speaker). As a result, the following clarifications need to be made:

a.  WTC remains high for pair work but not as high for group and whole class work, although this hypothesis is only confirmed in relation to virtual teaching.
b.  Student teachers might have expected to feel more comfortable interacting with other student teachers, despite course teaching modality and interlocutors' EFL proficiency. However, the results reveal that the least stimulating option is "someone I know a little" ("we are more willing to talk to a classmate we know better, perhaps because of embarrassment or fear of making a mistake at some point"; student teacher 4, personal communication, May 2020); i.e., student teachers prefer to interact with someone they already know or even with the teacher, despite his or her potential role as evaluator and not as facilitator for learning [29,58].
c.  Concerning EFL proficiency, the results confirm the hypothesis that WTC is higher when the interlocutor's knowledge is equal or lower than the student teacher him

or herself ("when I have a classmate with a language level proficiency higher than mine, more than embarrassment I feel afraid of not understanding his or her question, since I do not have the same level as he or she"; student teacher 10, personal communication, May 2020) both in the face-to-face and online course modality.

## 5. Conclusions

This research study confirms the fulfilment of the most significant overall didactic objective of the course: student teachers have become aware that it is mainly the use (and not the conscious learning) of EFL through oral communicative interaction what makes language acquisition possible [11]. Therefore, student teachers should also be ready to make use of the target language for most of the ordinary classroom routines soon.

On the other hand, one might now point out that the classroom (the physical setting where the teaching–learning process does traditionally take place) may be not the only scenery for the oral "exploration" (i.e., use) of EFL. Indeed, classrooms have usually been presented as the only appropriate contexts for activities that could require a relatively spontaneous use of the target language. However, the perceptions of student teachers reveal the idea that WTC in EFL has not been severely affected by the adaptation from face-to-face to virtual teaching with the only exception of grouping.

Instruction and syllabus adaptations derived from the COVID-19 in the field of EFL teaching constitute an invitation to extend oral communication activities beyond the traditional frontiers of the classroom setting. This measure affords blended learning a new opportunity to embrace not just individual productive tasks, autonomous learning goals, or writing activities that always bring a chance for the learner to review formal aspects of the target language, but also to support oral communication activities that promote the natural development of pupils' linguistic intuition

Overall, this study presents a context of initial teacher training that confirms positive effects on WTC from an ISLA-based methodological approach, at least for three elements of analysis (grouping, group member familiarity, and EFL proficiency) regardless of the teaching modality (face-to-face or virtual). However, the study also has limitations (low number of informants, the limited number of classes, the language proficiency of the student teachers, etc.) that may condition the results; thus, any claim of causality is to be taken with reserve. These limitations allow us to establish possible future lines of research, which would involve carrying out a longitudinal study that could refute or deny the results presented and replicating this work with student teachers from other degrees or even postgraduate courses in foreign language teaching.

**Author Contributions:** Conceptualization, J.L.E.-C. and F.Z.-M.; methodology, J.L.E.-C. and F.Z.-M.; validation, R.S.-C.; formal analysis, R.S.-C.; investigation, J.L.E.-C.; resources, J.L.E.-C.; data curation, R.S.-C.; writing—original draft preparation, J.L.E.-C. and F.Z.-M.; writing—review and editing, J.L.E.-C., F.Z.-M. and R.S.-C.; visualization, J.L.E.-C. and F.Z.-M.; supervision, J.L.E.-C. and F.Z.-M.; project administration, J.L.E.-C. All authors have read and agreed to the published version of the manuscript.

**Funding:** This research received no external funding.

**Institutional Review Board Statement:** This study was conducted in accordance with the Declaration of Helsinki and according to the Research Ethics Committee regulation from the *Universidad Autónoma de Madrid*, questionnaires are outside the scope of the application of Article 1.2. The present research is a noninterventional study that guarantees the anonymity of the participants, according to the Spanish Organic Law 3/2018, of 5 December, on Data Protection and Guarantee of Digital Rights.

**Informed Consent Statement:** Informed consent was obtained from all subjects involved in the study.

**Data Availability Statement:** Not applicable.

**Conflicts of Interest:** The authors declare no conflict of interest.

**Appendix A**

1. After attending the course (i.e., practical classes), what do you think teaching EFL in ECE consists of and how should it be carried out?
2. Which methodological elements worked on in the practical classes of the course do you think you will be able to use in your future EFL classes in ECE? Why?
3. To what extent is it possible to teach EFL in ECE through virtual classes because of the COVID-19, for example?
4. Concerning question 3, what alternatives would you propose if, as a future ECE teacher of EFL, a lockdown happens again?
5. The analysis of the qualitative data indicates that your WTC in English before the practical classes was higher for the virtual classes than for the face-to-face classes. How do you think you can overcome the feeling of embarrassment for your future work as an ECE teacher of EFL?
6. According to the analysis of the quantitative data on grouping, i.e., pairs, groups or whole class, the comparison between the first and the last face-to-face and virtual classes, respectively, shows positive opinions for both cases, except for one element (whole class). However, the qualitative data also show some variations if comparing face-to-face with virtual teaching in favour of the former, especially for one element (group work). Why do you think this is so, i.e., the difference between face-to-face and virtual teaching in relation to your willingness to speak in English?
7. Regarding the relationships between classmates in the classroom (i.e., a classmate I know very well; a classmate I know a little; or the teacher), the comparison between the first and the last face-to-face and virtual sessions, respectively, also shows positive opinions for the three options. Likewise, the comparison between face-to-face and virtual classes reveals better ratings for the first element (even though the option "with someone I know very well" had a better rating for virtual teaching). What do you think about these qualitative data on the relationships between classmates according to the distinction between face-to-face and virtual classes?
8. Among your opinions on WTC, there has also been a positive evolution as for interlocutor's EFL proficiency: her knowledge is equal to mine; higher than mine; or lower than mine. However, there exist some differences between face-to-face and virtual teaching, mainly regarding the option "higher than mine." What do you think about the difference between face-to-face and virtual teaching concerning your WTC as for the interlocutor's EFL proficiency?
9. How do you think that in the future some of the activities in relation to EFL teaching in ECE will be carried out virtually? List examples and give reasons for your answer.
10. Do you think that your proficiency in EFL has developed because of the practical classes of the course? To what extent? Why?
11. Do you think that your pedagogical skills as a future ECE teacher, mainly in the field of foreign languages, have developed because of the practical classes of the course? To what extent? Why?

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
