# Peer review of "Classroom-Situated Willingness to Communicate: Student Teachers of EFL in Spain"

_ejihpe, doi:10.3390/ejihpe13010007_

Round 1

Reviewer 1 Report

There are a couple of issues that need to be considered for the improvement of this manuscript:

(1) Although good context is provided as to where the study is taking place, at the beginning of the manuscript, there is no clear sense as to the bigger context and the importance of the phenomenon under examination. The manuscript lacks on literature that situates the phenomenon in the context of the scholarship within that research topic. 

(2) Currently, the organization of the manuscript reads and looks as if it is a proposal or a report on an instrument that has been developed and it validity and reliability is being reported. I think that there is a needs a rearrangement that will follow format that will allow for the manuscript to be clear for example: The Introduction does not have a clear literature review that can position this research in light of the scholarly literature within the relevant topic of research. The Method section needs to be clearly define as it pertains to who are the participants in the study, the instrument(s) used for the study, and the procedures followed for the study. Currently, the procedures reflect more of the instrument development and it is not clear how the data collection was done with participants. 

(3) It seems that issues with the Method section obscure the results and conclusion as one is not able to assess the worthiness of the results as it is not clear if the results are just instrument development relevant or if in reality the research questions have been answered. For example, in page 7, there is a focus on the analysis of the variables but no clear connection on how this answers the research questions. The current analysis seems to focus on item level descriptions but it is unclear what are the group differences. As a matter of fact, it is stated that the study is examining courses so what are the course differences and who are the students. 

Overall, I think that the study has merit but it need to be reorganize and refocus before publication. 

Author Response

Thank you very much for your selfless review and comments on our work. In attachments you can find a complete table with the changes and improvements made.

Reviewer 2 Report

The article presented is very pertinent and necessary. The starting point of the research is the learning model used for learning foreign languages. Different learning scenarios and their direct implications are developed. The research results confirm ISLA's methodological proposal, which seems very significant to us. We believe that it is essential to continue exploring and researching the different methods where language acquisition is through the promotion of interaction. 

As a recommendation, in the review of the literature, the different methods that have been the object of research in language learning could have been described. The research methodology seems appropriate in the quantitative field. Now, the triangulation of the data obtained, using a more qualitative approach, would have been of great interest.

Author Response

(The authors gave the same response as above.)
